# Allelopathic Effects of *Artemisia thuscula* and *Plocama pendula* on the Invasive Plant *Cenchrus setaceus* and Crops

**DOI:** 10.3390/plants14203159

**Published:** 2025-10-14

**Authors:** Ana Fuvel, Andreea Cosoveanu, Jorge Sopena Lasala, José Ramón Arévalo, Raimundo Cabrera

**Affiliations:** 1Agroecology and Environment Research Unit, Institut Supérieur d’Agriculture Rhône-Alpes (ISARA), 23 Rue Jean Baldassini, 69364 Lyon, France; anamfuvel@gmail.com; 2Department of Plant Sciences, Faculty of Biosciences, Norwegian University of Life Sciences, NO-1432 Ås, Norway; 3Department of Botany, Ecology and Plant Physiology, Faculty of Sciences, Section Biology, Universidad de La Laguna, 38206 La Laguna, Tenerife, Spain; jsopenal@ull.edu.es (J.S.L.); jarevalo@ull.edu.es (J.R.A.); rcabrera@ull.edu.es (R.C.); 4Gabinete de Estudios Ambientales (GEA), S.L.U, 38659 Tacoronte, Tenerife, Spain

**Keywords:** agroecology, biologicals, herbicide overuse, invasive species control, seed germination, aqueous extracts, allelopathic inhibition

## Abstract

*Cenchrus setaceus* is an alien invasive species with significant ecological impact on both natural ecosystems and agricultural areas across the Canary Islands. In this study, we evaluated the allelopathic effects of foliar lixiviates from two endemic species, *Artemisia thuscula* and *Plocama pendula*, on *Cenchrus setaceus* and a group of crop species to assess (i) germination inhibition of the invasive species and (ii) selectivity towards non-target crops. A preliminary trial tested undiluted and diluted forms (1%, 10%) of concentrated lixiviates prepared at a 1:3 (*w*:*v*) leaf-to-water ratio, using *C. setaceus* and *Lactuca sativa* under growth chamber conditions. In the validation trial, lixiviates prepared at a 1:6 (*w*:*v*) ratio were applied directly to *C. setaceus* and seven crops (*Zea mays*, *Allium cepa*, *Hordeum vulgare*, *L. sativa*, *Solanum lycopersicum*, *Brassica oleracea*, and *Raphanus sativus*) under both growth chamber and greenhouse conditions. Germination indices were calculated across assays, and plumule and radicle lengths were measured in growth chamber assays. In both trials, *C. setaceus* germination was inhibited by up to 60% by both ratios of lixiviates (Dunn *p* < 0.05), with reduced speed and seedling growth (plumule: −37.5%; radicle: −85%). Crop sensitivity varied: *A. cepa* and *H. vulgare* showed no significant inhibition; *B. oleracea* and *R. sativus* were affected by *P. pendula* (germination reduced 2.5–2.7×); and *Z. mays*, *L. sativa*, and *S. lycopersicum* exhibited delayed germination and reduced seedling growth under both treatments. These results support the selective use of native plant lixiviates for integrated management of *Cenchrus setaceus* in sensitive agroecosystems.

## 1. Introduction

Modern agriculture faces many challenges, such as biodiversity loss, pollution, climate change, and economic pressure, which threaten long-term sustainability. The growing awareness of these issues has driven research to take responsibility for finding solutions to reduce or eliminate the use of chemical inputs in agriculture and transition towards more sustainable and integrated systems.

Horticulture production plays a central role in European Food Systems, producing up to USD 100 billion, compared with about USD 40 billion for cereals [1]. However, current vegetable production systems heavily rely on chemical inputs, losing up to 54% of their production when avoiding the use of pesticides [2]. Weeds are, and will continue to be, the primary challenge for horticulture production in Europe due to their high adaptability to climate change and their invasive potential [3,4]. Nevertheless, according to Duke et al. [5], due to the extremely high cost of herbicide discovery and development, no herbicide with a new mode of action (MOA) has been released into the market for about two decades. This heavy reliance on chemicals with similar MOAs, which are usually based on a single active substance formulation, has led to 530 cases of herbicide-resistant weeds globally, in over 272 species [6,7]. Additionally, one of the predicted outcomes of climate change is a decrease in soil moisture due to drought, which contributes to a higher persistence of synthetic herbicides. This phenomenon may lead to herbicide carryover to following crops, especially in horticulture, increasingly endangering food quality and environmental safety [3]. The alarming situation regarding weed management has prompted the development of alternative solutions for maintaining productivity in agricultural systems [4]. This has sparked interest in allelopathy, particularly in how the effects of allelochemicals within agroecosystems can be beneficial and aid in creating alternative and sustainable strategies for weed management [8].

Beyond the general concept, multiple crop and horticultural systems already use allelopathic water extracts, mulches, or cover-crop residues with measurable weed suppression. Reviews synthesize how intercropping, rotation, cover crops, mulching, and aqueous plant extracts can be integrated operationally and report field-scale cases where allelopathy reduces herbicide inputs within Integrated Weed Management (IWM). For example, water extracts from sorghum, sunflower, and eucalyptus, individually or in combination, have suppressed germination and early growth of several weeds, and mixtures sometimes outperform single-species extracts in cereal systems. Assays on lettuce are also widely used to benchmark extract potency, underscoring method transferability to horticultural contexts. These lines of evidence support the feasibility of lixiviate-based approaches as pragmatic tools alongside mechanical control and cultural practices [4,9,10].

As allelopathic compounds play an important role in regulating plant communities in natural ecosystems, they can also be utilized as natural biodegradable herbicides [4]. Allelopathy finds application in field crops via intercropping, crop rotation, cover crops, mulching, and the use of allelopathic water lixiviates for weed management. Many real-world examples show how natural lixiviates can inhibit germination and initial plant development for invasive weed species. For instance, Tojic et al. (2025) [11] found that essential oils and aqueous extracts from *Artemisia absinthium* and *Artemisia vulgaris* significantly reduce germination and seedling growth of *Amaranthus retroflexus* and *Setaria viridis*. Another example is *Artemisia argyi*, whose aqueous extract was shown to suppress weed germination both in growth chamber and field trials [12].

Unlike chemical compounds, which kill non-target organisms beneficial for the agroecosystem, these biorational compounds tend to be more target-species specific and less likely to induce resistance [13]. In addition, as they are more rapidly degraded, generating low phytotoxic residue in crops, these are safer for farmers and greatly cheaper than bioherbicides [14]. Allelopathy shows a greater advantage against invasive weeds, as the species did not evolve together, thus preventing the possibility of developing resistance to these complex compounds [15]. This mechanism can be used with weeds, whether they are considered adventitious or alien species, as their development hinders the cultivated crops, in the case of agroecosystems, and the native plants, in the case of wild ecosystems. Allelopathy studies have encouraging results on the inhibition of germination, showing potential as a methodological approach in IWM in all the subtropical areas invaded by aggressive weeds. This approach is promising because it combines traditional knowledge and modern technology with an agroecological philosophy.

The Canary Islands, like many other territories in tropical and subtropical areas, have been seized by a highly invasive species, *Cenchrus setaceus* (Forssk.) Morrone (syn. *Pennisetum setaceum* (Forssk)), commonly known as fountain grass [16]. This plant species has been included since 2017 in the List of the Invasive Alien Species of Union concern [14,17], part of the 2020 Biodiversity Strategy of the European Union which implements the Biodiversity Targets of the Convention on Biological Diversity [15]. It can thrive in diverse environmental conditions and has escaped cultivation, invading a wide range of habitats, including France, Italy, Canary Islands, Hawaii, southern Africa, Democratic Republic of the Congo, Namibia, Fiji, southern continental United States, Australia, New Caledonia, and New Zealand [18]. This weed is particularly problematic in the Canary Islands and was already decreed in 2014 as “one of the most damaging invasive alien species for the natural and semi-natural environment of the Canary Islands” [19], as it has naturalized in several islands, from the coast to the Natural Reserves higher inland in the last 50 years [20]. It competes not only with other endemic and threatened species but also with agricultural crops [21].

As part of a broader strategy for the integrated control of *C. setaceus*, our research group (CIPEV—Integrated Control of Plant Pests and Diseases) initiated a study in 2021 to explore allelopathic solutions using two Canarian endemic species: *Artemisia thuscula* Cav. (Asteraceae) and *Plocama pendula* Ait. (Rubiaceae) against fountain grass. While other species from the same families have been largely documented for their rich chemical profiles, bioactivity, and applied uses in medicine and agriculture, *A. thuscula* and *P. pendula* remain less studied [22,23,24,25,26,27,28,29].

*Artemisia thuscula* produces artemisinin, a sesquiterpene lactone [22,29,30], and camphor [22], both being largely scrutinized molecules for their allelopathic potential. Multiple studies have proved the phytotoxicity of the artemisinin compound, which affects growth regulators at the DNA level, thus acting as a potent inhibitor of seed germination and plant growth on a wide range of species [4,7,31,32]. Similarly, Alanaz et al. [33] found inhibitory effects on germination using *Artemisia monosperma* Delile and *Artemisia judaica* L. extracts. Among other molecules, these species also produce camphor, which inhibits cell elongation, expansion, and division—processes which are essential for root growth [7,34,35]. Traditionally, *A. thuscula* has been used in Canarian agriculture as a tool against pests and diseases in the form of essential oils, aromatic smoke, and maceration [22,26,27]. These data illustrate mode-of-action diversity and a documented record in horticultural assays, strengthening the rationale to evaluate Canarian *Artemisia* resources beyond purely pharmacological uses.

Although *Plocama*, from the Rubiaceae genus, is comparatively under-documented, species from the same subfamily are known to be rich in anthraquinones and naphthohydroquinones, compounds frequently associated with antimicrobial and phytotoxic activities [36,37]. In *P. pendula*, root culture work has isolated a suite of novel and known anthraquinones (plocamanones, balonones, and derivatives), establishing a chemotypic basis for bioactivity and supporting the hypothesis that aqueous fractions can carry inhibitory constituents relevant to germination and early growth [38].

The objectives of this research were to determine the efficacy of the allelopathic effect of *A. thuscula* and *P. pendula* lixiviates against *C. setaceus* germination and to assess the levels of compatibility with crop species. Our preliminary assumption was that allelopathic compounds from *A. thuscula* and *P. pendula* would effectively inhibit the germination of *C. setaceus* even under less controlled conditions, such as in a greenhouse environment. We hypothesized that these compounds would exhibit low phytotoxicity towards common crops, given the long-standing ecological and anthropogenic coexistence of these species in both natural and agricultural landscapes. This hypothesis was supported by: (i) the historical use of *A. thuscula* in domestic and agricultural settings alongside cultivated species and (ii) the frequent occurrence of *P. pendula* in proximity to crops, suggesting potential compatibility.

## 2. Results

### 2.1. Preliminary Trial

In the preliminary assay, *Cenchrus setaceus* seeds were exposed to lixiviates of *Artemisia thuscula* and *Plocama pendula* at both concentrated and diluted levels. Overall, germination was strongly inhibited under concentrated treatments (100%), particularly with *A. thuscula*, which reduced germination success by more than half (Games-Howell, *p* < 0.05; Table 1). In contrast, diluted extracts (1% and 10%) had a minimal impact on germination percentage, although moderate delays in germination speed were observed at 10%, especially under *A. thuscula* (Dunn, *p* < 0.05).

The Germination Index (GI) and Mean Germination Time (MGT, days) confirmed these trends: the GI dropped by 75–80% under concentrated extracts and by ~25% under 10% dilution, while MGT doubled in the latter case, indicating a significant slowdown in germination dynamics (Dunn, *p* < 0.05).

Seedling development was also markedly impaired in the presence of lixiviates. Radicle growth was nearly stopped by *P. pendula* even at lower concentrations, whereas plumule length was more affected by *A. thuscula*. These patterns were consistent across both concentration levels, with higher doses leading to stronger inhibitory effects (Dunn, *p* < 0.05).

*Lactuca sativa*, included as a crop reference, showed a similar but milder response. While germination percentage remained unaffected, germination speed and seedling elongation, particularly radicle development, were significantly reduced under both treatments, more so with *A. thuscula* (*p* < 0.05).

### 2.2. Validation Experiment: Growth Chamber Assays

#### 2.2.1. Germination Dynamics

*Artemisia thuscula* (treatment A) and *Plocama pendula* (treatment B) lixiviates showed a tendency to decrease seed sprouting for both *C. setaceus* and the crop species except *H. vulgare*, with a higher inhibitory effect of treatment A (Appendix A). Germination dynamics showed a slight delay in the first sprouting events when lixiviates were applied for all species except *H. vulgare* and *Z. mays*. Yet, germination variability between plates showed fluctuation of germination trends for *H. vulgare*, *Z. mays*, and *A. cepa* (Appendix A), which was translated as a lack of homogeneous response on the calculated indices.

Treatment effects, either one or both compared with the control group, showed significantly different values for the FPG, MGT, and GI indices for all species except *H. vulgare* (Dunn, *p* < 0.05). The FPG index was significantly lower for the seeds under treatment A and treatment B lixiviates compared with the control (Dunn, *p* < 0.05). For treatment A, the average value of germinated seeds decreased from 1.7× for large-seed species (e.g., *Z. mays*) up to 168× for small-seed species (e.g., *S. lycopersicum*). For treatment B, effects ranged from 1.3× (*Z. mays*) to 24.6× (*B. oleracea*). Similarly, MGT values were higher considering all species combined except *Z. mays*, *H. vulgare*, and *R. sativus*, increasing values on average by 1.3 and 1.9 days for treatment A and B, respectively, indicating a significantly slower germination speed for seed populations compared with the control (Games-Howell, *p* < 0.05). *Z. mays* and *R. sativus* were only affected by treatment B, slowing the germination of the seed population by 0.5 and 3.5 days, respectively (Games-Howell, *p* < 0.05). The GI average values for both treatments represented 30% of the control group, indicating a slower speed and lower success of the sprouts (Dunn, *p* < 0.05; Figure 1).

Species responded differently to treatments when evaluating the first and last day of germination (FDG and LDG). Treatment A showed high inhibition on *S. lycopersicum*, so much that group differences for the FDG and LDG could not be calculated. When considering the FDG index, both lixiviates induced similar delays of the start of the germination events for *B. oleracea*, *L. sativa*, *R. sativus*, and *C. setaceus* compared with the control (Dunn, *p* < 0.05; Appendix A). *A. cepa* was only significantly affected by treatment A, having its germination delayed by 1.8 (±1.7) days, while treatment B delayed the first sprouting of *S. lycopersicum* by 4.7 (±2.88) days (Dunn, *p* < 0.05, Appendix A). The presence of treatments induced milder effects on the last day of germination (LDG) across species, with only treatment B delaying the last day of germination event in *L. sativa* by an average of 2 days (±1.67) compared with the control (Dunn, *p* < 0.05, Appendix A).

#### 2.2.2. Seedling Viability—Plumule and Radicle Lengths

Plumule (PL) and radicle (RL) lengths (mm) were significantly hindered when in the presence of lixiviates for all species (Dunn, *p* < 0.05). On average, PL length ranged between 19.3× and 1.8× less when comparing treatment A with the control groups. Similarly, treatment B diminished the PL length by between 11.9× and 1.4×, depending on the species (Table 2). RL length was reduced by 1.8× to 4.8× under treatment A, and by 1.7× to 10.3× under treatment B, depending on the species. *C. setaceus*, *L. sativa*, and *A. cepa* were the only species where the radicle development was more hindered than the plumule (Table 2). Both *Z. mays* and *L. sativa* were affected twice as much by *A. thuscula* than by *P. pendula*.

### 2.3. Validation Experiment: Greenhouse Assays

*A. thuscula* and *P. pendula* lixiviates tend to reduce the number of seed sprouts for all species except *H. vulgare* and *A. cepa* (Appendix A). Compared with the growth chamber, *A. cepa*, *B. oleracea*, *L. sativa*, and *R. sativus* responded differently in the presence of treatment B, with the latter inhibiting germination more than treatment A. In the presence of both treatments, *L. sativa*, *Z. mays*, *C. setaceus*, and *S. lycopersicum* had a slight delay of the first day of germination event. All species showed a high variability of germination count values among replicates (Appendix A) compared with the growth chamber assays. This indicates higher heterogeneity among responses of individuals to the presence of treatments.

*Z. mays*, *L. sativa*, *S. lycopersicum*, and *C. setaceus* were similarly affected by both lixiviates, with final percentage of germination values inhibited by 1.3× to 2.5× (Dunn, *p* < 0.05). The FPG values of *B. oleracea* and *R. sativus* were only significantly lowered by treatment B, by 2.5× and 2.7×, respectively (Dunn, *p* < 0.05, Appendix A). *A. cepa* and *H. vulgare* showed no significant differences between treatments and the control (Dunn, *p* > 0.05, Figure 2). MGT indicated a slower germination speed for all species except *A. cepa* and for *H. vulgare* in the presence of both lixiviates. Large-seed species responded differently than small-seed species, with the former being delayed by an average of 0.3 days and the latter by 3 days (Appendix A). According to the GI values, *Z. mays*, *L. sativa*, and *C. setaceus* were equally affected by both lixiviates, decreasing their rate of germination and the number of events compared with the control group by 50%, 63%, and 71%, respectively (Dunn, *p* < 0.05). Compared with the control, in the presence of treatment B, *B. oleracea* and *R. sativus* had their GI values decreased by 65% and 78%, respectively, and by 55% and 25%, respectively, in the presence of treatment A (Dunn, *p* < 0.05). Contrarily, *S. lycopersicum* was 10% more affected by treatment A than by treatment B, having its GI values lowered by 81% and 70%, respectively, compared with the control (Dunn, *p* < 0.05; Figure 2, Appendix A).

All species except *A. cepa* and *H. vulgare* had a delayed start of sprouting when in the presence of lixiviates. For *Z. mays*, *B. oleracea*, *R. sativus*, and *C. setaceus*, both lixiviates induced 1–2 days of delay for the FDG events compared with the control (Dunn, *p* < 0.05; Appendix A). Compared with the control, in the presence of treatment A, *L. sativa* and *S. lycopersicum* had their FDG events delayed by 3 and 4 days, respectively, whereas in the presence of treatment B, the first events were delayed by 2 and 3 days (Dunn, *p* < 0.05; Appendix A). When considering the LDG index, species presented higher sensitivity to the lixiviates under greenhouse conditions than in the growth chamber assays (Appendix A). Compared with the control group, *L. sativa* and *S. lycopersicum* had their the LDG events delayed by both lixiviates by 1–2 and 3 days, respectively (Dunn, *p* < 0.05, Appendix A). Likewise, *Z. mays* showed a similar trend but only in the presence of treatment B (Dunn, *p* < 0.05). The LDG events of *R. sativus* and *C. setaceus* were delayed in the presence of treatment B compared with the control (Dunn, *p* < 0.05, Appendix A). *A. cepa* was the only species to present an early LDG event in the presence of treatment A compared with the control (Dunn, *p* < 0.05, Appendix A).Combined LDG and FDG values indicate only a temporal shift in germination events for both treatments (i.e., *A. cepa*, *H. vulgare*, *L. sativa*, and *Z. mays*).

## 3. Discussion

This study demonstrates that aqueous lixiviates from the Canary Islands endemics, *Artemisia thuscula* and *Plocama pendula*, strongly inhibit germination and early growth of the invasive grass *Cenchrus setaceus* while exerting variable effects on crop species. On average, germination of *C. setaceus* was reduced by nearly 60% under both growth chamber and greenhouse conditions, with substantial suppression of plumule and radicle development. These results confirm that allelopathic extracts derived from native plants can contribute to the management of aggressive invaders in fragile island ecosystems.

Growth chamber assays highlighted the dose-dependent and treatment-specific inhibitory effects of both endemic species. *C. setaceus* and *L. sativa* displayed distinct species-specific responses, with higher concentrations (1:3, *w*:*v*, leaves:water, undiluted) exerting consistently stronger inhibition than diluted forms (1:3 at 10% and 1%, and 1:6 undiluted). This dose–response pattern suggests that phytotoxic sensitivity is not only linked to allelochemical presence but also to species-specific thresholds for uptake or detoxification [39]. Radicle growth was more strongly inhibited than plumule elongation, consistent with the well-documented sensitivity of root meristems to allelochemicals that disrupt cell division and elongation [34].

When compared with chemical graminicides, the efficacy of the lixiviates was of the same order. Catanzaro et al. [40] reported similar values of germination inhibition and young plant mortality rates of *C. setaceus* (60% and 55%, respectively) under greenhouse conditions. Likewise, Duke et al. [31] demonstrated that *Artemisia annua* L. acted comparably to synthetic herbicides like cinmethylin. These parallels reinforce the potential of native allelopathic resources as realistic alternatives for weed suppression.

From a practical perspective, the observed efficacy against *C. setaceus* indicates that lixiviates or biomass of *A. thuscula* and *P. pendula* could be incorporated into IWM. Their use would complement mechanical removal and reduce reliance on herbicides, which is especially relevant in the Canary Islands where chemical inputs are already high and environmental vulnerability is pronounced.

Several application routes can be proposed. Hedgerows of the endemic species may act as buffer zones between crops and natural areas, releasing allelochemicals via litter or rainfall wash-off while simultaneously functioning as windbreakers and habitats for beneficial arthropods. Plant biomass could be applied as mulch or green manure to extend suppressive effects, improve soil fertility, and retain moisture. These strategies are consistent with recommendations from recent reviews emphasizing the integration of allelopathic plants, cover crops, and mulches into diversified weed management systems [4,41]. In addition, locally adapted endemics could support biodiversity and open opportunities for community-based nursery production, linking weed management with both ecological and socio-economic benefits.

Species grouped by sensitivity (Appendix A) fall into two broad clusters: sensitive vs. tolerant; tolerant species tend to be those with larger seeds or more robust seed coats (e.g., *Z. mays* and *H. vulgare*), indicating that seed traits mediate allelochemical impact. Duke et al. [31] observed that artemisinin exerts greater inhibition on small-seed species than large-seed species. Due to the large size of the seed, a higher volume of water is retained; therefore, a dilution effect of the absorbed solution of the lixiviate may be also responsible. As many studies show that a higher concentration of extracts is equal to greater inhibition, a pro rata of lixiviate needed, depending on seed size, could be calculated [31,32,33]. This phenomenon could also be due to seed coat protection. Large seeds often have thicker and more robust seed coats, which can act as a physical barrier to allelopathic compounds. This protection can reduce the exposure of the embryo to harmful molecules, allowing the seed to germinate and establish more effectively [42].

Even though some tested crops showed sensitivity, the applicability of this approach should not be disregarded. Non-target damage is also common with synthetic herbicides through spray drift or overspray [43,44]. To contrast the predicted 54% yield reduction in horticulture without herbicide use [2], we compared the halved final germination values of the control groups with those of the crops exposed to lixiviates (Appendix A). In the growth chamber assays, all species except *Z. mays* and *H. vulgare* showed equal or better FPG under treatment than the halved control values. Under greenhouse conditions, treatment A increased FPG in *A. cepa*, *B. oleracea*, *L. sativa*, and *R. sativus* while treatment B increased FPG in *S. lycopersicum* and *Z. mays* (Dunn, *p* < 0.05). Such comparisons demonstrate that selectivity, though not absolute, can be exploited within IWM schemes.

Current control of *C. setaceus* relies mainly on manual removal or chemical herbicides, particularly glyphosate or hexazinone [45,46]. Both approaches face serious limitations: manual removal is labor-intensive and costly, while herbicides bind strongly to soils, persist in water, and damage native flora [46]. In landscapes with rugged topography or extensive infestations, such as cliffs or ravines of the Canary Islands, neither method has proven successful [45]. The Canary Islands already record the highest pesticide use and agrochemical residues in [47,48]. Against this backdrop, allelopathic methods emerge as a promising ecological alternative.

The reintroduction of *A. thuscula* and *P. pendula* into agroecosystems would deliver additional benefits beyond direct weed suppression. On an environmental level, increased biodiversity reduces opportunities for adventitious infestation [4,15,49]. Hedgerows act as windbreakers in areas exposed to strong trade winds [50] and can shelter beneficial species such as *Orius* spp., a natural predator of thrips, which use *P. pendula* as refuge [51]. Mulch and green manure practices are known to improve soil moisture and fertility while limiting weed establishment [4,44]. Economically, endemic plant propagation could support local nurseries, while socially it could reconnect farmers with traditional agricultural knowledge and promote conservation awareness [22,26,52]. In natural reserves where *C. setaceus* has built persistent seed banks lasting up to seven years [53], planting endemics may assist restoration by reducing invasive pressure [54] and mitigating fire risk, for which this grass is a major factor [55,56].

Nevertheless, several limitations must be acknowledged. First, all assays were performed under controlled conditions; field translation remains uncertain. Responses varied among species, with *C. setaceus* showing stable inhibition across environments while *S. lycopersicum* displayed up to 99% increased resistance under greenhouse conditions with *A. thuscula* lixiviates. Different responses between environments may be explained by soil heterogeneity, microbial activity, or climatic variability [14,49,57]. Second, allelochemical persistence is typically short: compounds degrade due to light, moisture, and microbial activity, necessitating repeated application or continuous biomass inputs [58]. Third, crop compatibility is not universal, as lettuce and tomato proved sensitive. Finally, scaling-up entails challenges in extract standardization, dosage regulation, and environmental safety [59].

Future studies should therefore prioritise field trials to assess biomass production, competitive suppression, and durability of effects under realistic agronomic and ecological conditions. Research on delivery methods—such as dry biomass, aqueous extracts, or incorporation as green manure—will help balance efficacy and selectivity. Longitudinal monitoring of persistence and non-target effects is required, along with cost–benefit comparisons against mechanical removal and herbicide application.

Allelopathy must be considered within the broader ecological interactions of agroecosystems [14,49]. Field validation should address not only the persistence of inhibitory effects but also potential competition for resources or shading when endemics are used as hedgerows. In this way, hedgerows, mulches, and other practices can be integrated as complementary tools within IWM, contributing both to biodiversity and to sustainable crop production.

In conclusion, *A. thuscula* and *P. pendula* show promise as bioherbicidal agents against *C. setaceus*. While challenges remain regarding scale-up and crop compatibility, the integration of these endemic species into diversified weed management strategies could reduce herbicide reliance, support conservation goals, and promote sustainable agriculture in island ecosystems.

## 4. Materials and Methods

### 4.1. Collection and Handling of Plant Samples

Leaves from *A. thuscula* and *P. pendula* were collected from individuals at the vegetative state, in Arico, located in the south of the island of Tenerife, at all time points when necessary to prepare the lixiviates.

We selected for the preliminary trial (growth chamber) a single species, *L. sativa*, to be used as a sensitive plant model and for the validation trial (both growth chamber and greenhouse) seven crop target plant species using commercial seeds in order to evaluate germination trends. Monocotyledons were represented by *Zea mays* L. (var. Golden Bantam), *Hordeum vulgare* L. (var. Kamalamai), and *Allium cepa* L. (var. Amposta); for the dicotyledons, we selected *Lactuca sativa* L. (var. Bionda Ortolani), *Solanum lycopersicum* L. (var. Beefsteak), *Brassica oleracea* L. (var. Brunswick), and *Raphanus sativus* L. (var. Scarlet round) *Cenchrus setaceus* panicles were collected from healthy individuals bearing numerous inflorescences, sampled repeatedly from the same plant communities along the roadsides of San Cristóbal de La Laguna (Tenerife, Canary Islands), during both preliminary assays (September and December 2021) and validation assays (September 2023). Seeds were obtained from hermaphrodite flowers for subsequent assays.

*C. setaceus* plant material was handled under the authorization to access Spanish plant genetic resources from wild taxa (Ministerio para la Transición Ecológica y Reto Demográfico) and collected under the authorizations no. EEI-006/2019 (no. 1959553) and no. ESNC54, issued by the Ministry of Ecological Transition, Fight against Climate Change, and Territorial Policy of the Autonomous Government of Canary Islands (Dirección General de Lucha contra el Cambio Climático y Medio Ambiente del Gobierno de Canarias). The collection and manipulation of the seeds adhered to the guidelines set forth by the Nagoya Protocol, ensuring compliance with all relevant regulations.

### 4.2. Preliminary and Validation Experiments

The study was structured into two sequential trials: a preliminary and a validation one. Each trial included multiple assays designed to assess the effects of foliar lixiviates from two endemic plant species, *Artemisia thuscula* and *Plocama pendula*, on the germination and seedling development of *Cenchrus setaceus* and several crop species. The trials differed in three key aspects: (i) the concentration of lixiviates, (ii) the target species, and (iii) the inclusion of greenhouse assays, which were exclusive to the validation trial.

The preliminary *trial* (small-scale exploratory) focused on *C. setaceus* as the main target species. After initial observations, *Lactuca sativa* was added as a model species for sensitivity evaluation. Foliar lixiviates were prepared at a 1:3 (*w*:*v*) fresh leaf-to-water ratio. Fresh leaves of *A. thuscula* and *P. pendula* were collected, cut into fragments, and immersed in ultrapure water on the same day. The suspensions were left overnight in darkness at room temperature to allow passive leaching of compounds. Solutions were then centrifuged (4000 rpm, 5 min) and filtered through a 0.22 μm membrane using a Büchner funnel under vacuum (300 mbar) under sterile conditions. Filtrates were stored at 4 °C and used within 24 h. This protocol follows ecologically oriented allelopathy methods that aim to simulate rainfall-induced leaching or short-term wash-off from foliage [60,61]. To explore the dose–response relationship, two dilutions (10% and 1%) of the lixiviates were also tested on both species. Germination was monitored under growth chamber conditions for 14 days in *C. setaceus* and 8 days in *L. sativa* based on species-specific germination dynamics.

To simulate more realistic field conditions, the *validation trial* used a 1:6 (*w*:*v*) ratio for lixiviate preparation. The same fresh-leaf extraction method was followed: soaking overnight in ultrapure water and then filtering through Whatmann No. 1 filter paper. For growth chamber assays, a second filtration step was added using a borosilicate glass-fritted funnel (FU23-047-001). All lixiviates were used within 24 h and stored at 4 °C. Although pH and electrical conductivity of the lixiviates were not measured directly, all solutions were prepared using ultrapure water, with consistent extraction conditions across repetitions and treatments. Eight species were tested: the invasive grass *C. setaceus* and seven crop species (*Z. mays*, *H. vulgare*, *A. cepa*, *L. sativa*, *R. sativus*, *B. oleracea*, and *S. lycopersicum*), under both growth chamber and greenhouse conditions. Due to varying germination times among species, these were grouped into: Group 1 (*Z. mays*, *L. sativa*, *R. sativus*, and *B. oleracea*) monitored for 8 days and Group 2 (*A. cepa*, *S. lycopersicum*, *H. vulgare*, and *C. setaceus*) monitored for 14 days.

The *C. setaceus* seeds used in the preliminary trial were collected at two time points in the same year (September and December), whereas the seeds for the validation trial were collected two years later, also in September. The seeds were manually extracted by dissecting each fascicle and isolating the hermaphroditic flower, removing lemma and palea to obtain clean caryopses. All seeds were air-dried and stored under ambient laboratory conditions until use. This labour-intensive step ensured seed purity. The seeds were air-dried and stored in airtight glass containers with silica gel at room temperature. Prior to use, the seeds were surface-sterilized in 0.2% NaOCl solution for 10 min under vortex agitation (1800 rpm, 3 pulses), followed by three rinses with sterilized ultrapure H_2_O and then air-dried in a fume hood.

In both trials, germination was assessed daily in growth chambers using 100 seeds per species–treatment combination. The seeds were placed on double-filter paper in sterile Petri dishes (9 cm diameter, except for *Z. mays*, which used 15 cm dishes), with 20 seeds per dish and three technical replicates. Filter papers were moistened with 2.5 mL (or 3.5 mL for large dishes) of either treatment A or B or of the control solution (ultrapure water). For each assay, germination assessments were made daily across 100 seeds per species–treatment combination. In order to assess seedling development, the seedlings were scanned at the end of the assay, and 60 randomly selected individuals per group were analysed using Image J v1.53 to measure plumule (PL) and radicle (RL) lengths. In the preliminary trial, all available seedlings (*n* = 100) were measured.

In the greenhouse, 99 seeds per species–treatment combination were sown, with 3 seeds per seedbed. The substrate used was a commercial mix of black peat, perlite, Agrosil, and nutrients (COMPO *Sana^®^ Universal*) with the following properties: organic matter 96%, pH 5–6.5, N:P:K = 200–450:200–500:300–550 mg L^−1^, and salt content < 3 g L^−1^. At the beginning of the assay, the substrate of each seedbed was saturated with 30 mL of the lixiviate or ultrapure H_2_O. Daily mist irrigation maintained substrate moisture. Germination was assessed daily using all seeds (*n* = 99) per species–treatment combination.

Each assay (growth chamber and greenhouse) was performed twice as independent biological repetitions using the same protocol, lixiviates, and incubation conditions. Growth chamber conditions were maintained at 25/18 °C light/dark), with a 12:12 photoperiod, 55–60% relative humidity, and a daily photon flux density of 50 μmol m^−2^ s^−1^. In the greenhouse, minor temperature variation was recorded due to seasonal progression (February to April), but both repetitions occurred within the wet season, minimizing environmental variability. Average monthly temperatures were: 21.2 ± 18.1 °C (February), 22.1 ± 12.9 °C (March), and 25.3 ± 15.1 °C (April).

### 4.3. Statistical Analysis

To evaluate the effect of the lixiviates on germination and the modulation of the germination process, we calculated germination indices: (i) Final Percentage Germination (FPG) expressed as the percentage of germinated seeds at the end of the trial [62]; (ii) Mean Germination Time (MGT) estimating the average germination speed [63]; (iii) the Germination Index (GI) combining germination percentage and rate [64]; and (iv) the First Day of Germination (FDG) and the Last Day of Germination (LDG) recording the first and last germination events [65]. Additionally, for the germinated seeds from the growth chamber assays, the plumule (PL) and radicle (RL) lengths (mm) were measured.

Because ANOVA model residuals did not meet normality and homoscedasticity assumptions, we applied the non-parametric Kruskal–Wallis test (*kruskal**.test* function, “stats” package) [66]. When significant effects were detected, we conducted pairwise comparisons using the post hoc Dunn’s test with Bonferroni correction (*dunn.test* function, “dunn.test” package) [67]. Where data distribution and variance allowed, we performed Welch’s ANOVA test (*aov* function) [66] followed by Games-Howell post hoc comparisons (*games_howell_test* function, “userfriendlyscience” package) [68].

For time-related indices (MGT, FDG, LDG), seeds that failed to germinate were assigned a “NA” value instead of a “0” value, as this would artificially imply immediate germination. Similarly, non-germinated seeds were excluded from plumule and radicle measurements to ensure valid comparisons between seedlings.In cases where germination was extremely limited or absent under certain treatments (e.g., *B. oleracea*, *R. sativus*, and *S. lycopersicum* in the validation trial, and *L. sativa* and *C. setaceus* in the preliminary trial at 100% lixiviates), analyses of plumules and radicles could not be performed.

To explore species sensitivity profiles to lixiviates, we clustered the FPG and GI values using the k-means algorithm (*kmeans* function, “stats” package) [66]. The number of clusters was manually selected based on grouped species–treatment data.

All tests used a significance threshold of 0.05. Data were structured and reshaped using “dplyr” [69] and “tidyr” [70], and visualizations were created using “ggplot2” [71]. All statistical analyses were performed using R v.4.4.1 [66] within RStudio v.2022.02. [72].

## 5. Conclusions

This study shows that aqueous lixiviates from the Canary Island endemics *Artemisia thuscula* and *Plocama pendula* strongly inhibit germination and early growth of the invasive grass *Cenchrus setaceus*, while affecting crop species selectively. These findings highlight their potential as locally sourced bioherbicidal tools.

In practical terms, the use of endemic species in hedgerows, mulches, or lixiviate-based treatments could provide complementary options within IWM, reducing reliance on synthetic herbicides in sensitive island ecosystems. Future research should focus on field validation under agronomic conditions, chemical profiling of active compounds, and optimisation of application methods to ensure efficacy, persistence, and crop compatibility.

## Figures and Tables

**Figure 1 plants-14-03159-f001:**
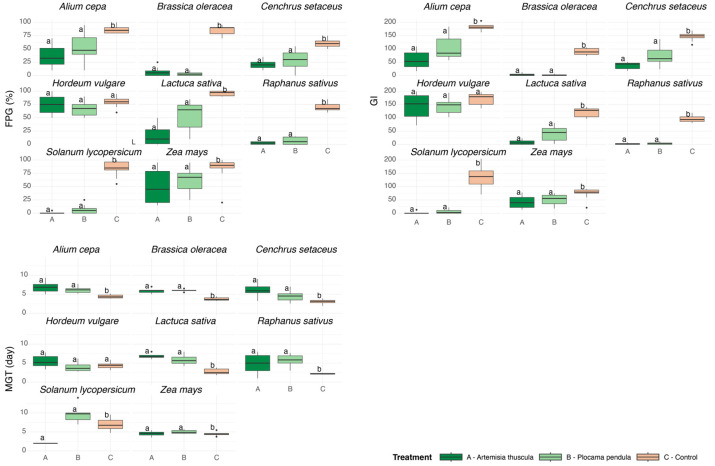
Germination response of species to foliar lixiviates of *Artemisia thuscula* and *Plocama pendula* compared with the untreated controls under growth chamber conditions. Final Percentage of Germination (FPG, %), the Germination Index (GI), and Mean Germination Time (MGT, days) are displayed as boxplots (left to right, top to bottom). Boxplots represent the median, 25th, and 75th percentiles and the range of nearby values; dots indicate outliers. Different letters above the boxes indicate significant differences between treatments (*p* < 0.05).

**Figure 2 plants-14-03159-f002:**
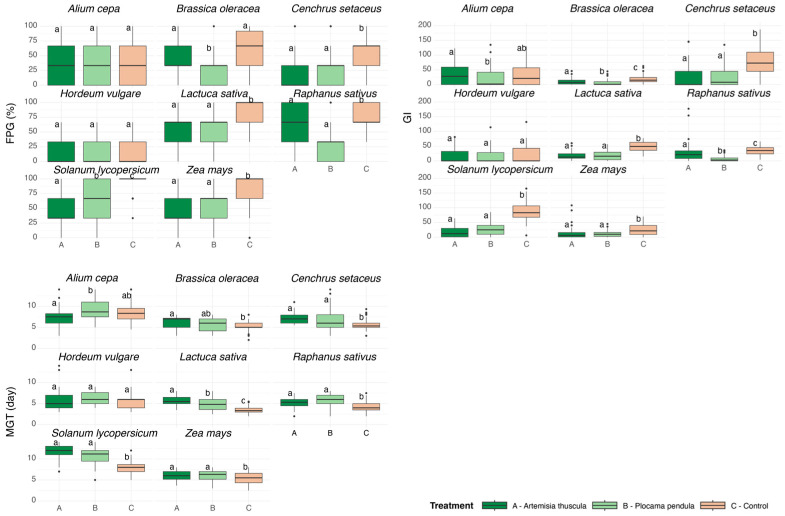
Germination response of species to foliar lixiviates of *Artemisia thuscula* and *Plocama pendula* compared with the untreated controls under greenhouse conditions. Final Percentage of Germination (FPG, %), the Germination Index (GI), and Mean Germination Time (MGT, days) are displayed as boxplots (left to right, top to bottom). Boxplots represent the median, 25th, and 75th percentiles and the range of nearby values; dots indicate outliers. Different letters above the boxes indicate significant differences between treatments (*p* < 0.05).

**Table 1 plants-14-03159-t001:** Germination dynamics and seedling lengths of *Cenchrus setaceus* and *Lactuca sativa* under different treatments.

Species	T	[Conc]	FPG (%)	GI (Index)	MGT (Days)	PL (mm)	RL (mm)
*C. setaceus*	A	1	63 ± 12.94	128.6 ± 37.6 A	4.21 ± 1.17	54.1 ± 23.6	73.15 ± 27.86 aA
	A	10	63 ± 16.81	100 ± 31.38 b	7.14 ± 0.76 bA	48.71 ± 18.19	24.73 ± 13.97 bA
	A	100	0 ± 0 b	0 ± 0 b	-	-	-
	B	1	80 ± 6.12	190.2 ± 19.64 B	3.13 ± 0.42	48.77 ± 20.27	51.51 ± 23.02 bB
	B	10	60 ± 22.08	138.6 ± 51.66	3.44 ± 0.49 B	44.42 ± 10.23 b	8.41 ± 6.38 bB
	B	100	4 ± 4.18 b	9 ± 8.77 b	2 ± 2.12	-	-
	C	0	71 ± 9.62 a	169.8 ± 26.58 a	3.07 ± 0.35 a	49.77 ± 19.06 a	69.47 ± 25.5 a
*L. sativa*	A	1	95 ± 5	143.6 ± 14.57 bA	1.19 ± 0.43	13.46 ± 3.66 b	81.22 ± 27.73 b
	A	10	93 ± 4.47 b	110.4 ± 13.87 bA	3.08 ± 0.47 bA	10.32 ± 3.01 A	15.52 ± 8.45 bA
	A	100	0 ± 0 b	0 ± 0 b	0 ± 0 b	-	-
	B	1	100 ± 0	166.4 ± 17.14 B	1.08 ± 0.06	12.59 ± 3.02 b	87.46 ± 25.36 b
	B	10	96 ± 4.18	146.6 ± 8.62 bB	1.37 ± 0.14 bB	12.05 ± 3.59 bB	22.4 ± 5.26 bB
	B	100	0 ± 0 b	0 ± 0 b	0 ± 0 b	-	-
	C	0	100 ± 0 a	157.8 ± 2.28 a	1.11 ± 0.11 a	10.24 ± 2.86 a	61.72 ± 26.94 a

FPG—Final Percentage of Germination, GI—Germination Index, MGT—Mean Germination Time(days), PL—plumule length (mm), and RL—radicle length (mm). Values are expressed as mean ± standard deviation. Treatments (T): A—*Artemisia thuscula*, B—*Plocama pendula*, and C—control. Concentrations (Conc): 1%, 10%, and 100% of the lixiviate (A or B). Different lowercase letters indicate significant differences (*p* < 0.05) between treatments A and B compared with the control. Different uppercase letters indicate significant differences between treatments A and B at the same concentration. “-“ indicates values that could not be calculated due to the absence of germination (FPG = 0), making the GI, MGT, PL, and RL non-measurable.

**Table 2 plants-14-03159-t002:** Mean ± SD values of plumule and radicle lengths and the number of non-germinated seeds per treatment (A = *A. thuscula*, B = *P. pendula,* C = control).

Species	PL (mm)			RL (mm)			Number of Non-Germinated Seeds		
	A	B	C	A	B	C	A	B	C
*A. cepa*	7.45 ± 5.2	6.34 ± 5.02	17.37 ± 12.65	3.25 ± 1.84	3.24 ± 2.19	12.45 ± 9.38	80	54	15
*C. setaceus*	10.22 ± 8.60	13.35 ± 10.72	18.74 ± 12.26	6.39 ± 6.09	2.93 ± 2.80	30.26 ± 18.19	96	69	49
*H. vulgare*	2.22 ± 9.99	1.57 ± 4.05	9.7 ± 23.31	3.78 ± 6.34	2.98 ± 2.22	8.57 ± 16.85	28	48	25
*L. sativa*	2.36 ± 3.42	4.68 ± 3.16	9.98 ± 2.66	7.43 ± 5.41	4.89 ± 2.54	32.87 ± 16.85	98	42	6
*Z. mays*	0.39 ± 1.3	0.63 ± 1.54	7.46 ± 13.85	12.02 ± 14.03	12.93 ± 5.28	21.8 ± 16.51	50	34	9

## Data Availability

The datasets generated and analysed during the current study are available from the corresponding author on reasonable request.

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
