# Peer review of "Allelopathic Effects of Artemisia thuscula and Plocama pendula on the Invasive Plant Cenchrus setaceus and Crops"

_plants, 2025, doi:10.3390/plants14203159_

Round 1
Reviewer 1 Report
Comments and Suggestions for Authors
The topic is interesting and addresses a relevant issue in weed management using ecological approaches. However, there are several major and minor issues that need to be addressed before the manuscript can be considered for publication. Below are my detailed comments:
Major Comments:
Title and Scope: The term "Horticultural Crops" is inaccurate since Zea mays and Hordeum vulgare are field crops rather than horticultural species. Suggest revising to "Agricultural Crops" or "Selected Crops" to avoid confusion.
Abstract: The abstract overly emphasizes results on Cenchrus setaceus while neglecting quantitative findings on other species. Please include key results for non-target species.
Keywords: Current keyword are too broad. Recommend replacing with more specific terms such as "allelopathic inhibition", "aqueous extracts", "seed germination", and "invasive species control".
Introduction: The introduction lacks background on related species with known allelopathic effects. Adding this would strengthen the rationale for studying A. thuscula and P. pendula.
Results Section: Avoid mechanically repeating numerical values from figures/tables. Instead, synthesize trends. Merge results from the two trials and discuss combined effects of high and low concentrations thematically.
Data Presentation: Replace raw data with standardized metrics: germination rate, germination energy, vigor index, plumule length, and radicle length. Move raw data to supplementary materials if necessary.
Figure 5: The cluster analysis lacks biological interpretation and appears redundant. Recommend removal unless deeper ecological insights are provided.
Discussion: Delete lengthy repetitions of results and speculative content on molecular mechanisms. Focus on practical implications of the findings (e.g., potential for integrated weed management) and limitations.
Methodology: The protocol for preparing extracts (fresh leaves chopped and soaked in water) deviates from classical allelopathy studies (typically using dried powdered material). Justify this approach with references and explain how it affects compound extraction efficiency.
Statistical Analysis: The description of statistical methods is confusing. Clarify how missing data or zeros were handled. Specify post-hoc tests used for pairwise comparisons.
Minor Comments:
Latin Names: Italicize all scientific names consistently in the main text and figures (e.g., Cenchrus setaceus, Artemisia thuscula).
Table and Figure Captions: Captions should describe content directly (e.g., "Germination rates under different treatments") rather than analytical methods (e.g., "Cluster analysis based on FPG and GI").
Section Headings: Avoid informal headings like "Up-Scaled Trial". Use standard academic phrasing (e.g., "Greenhouse Validation Experiment").
Germination Conditions: Specify whether all species were germinated under identical conditions (temperature/light). If not, discuss potential impacts on results.
Conclusion: Expand the conclusion to highlight practical applications and future work.
Comments on the Quality of English Language
The manuscript requires extensive language polishing to address inconsistent tense usage, awkward phrasing, and grammatical errors for improved academic clarity.
Author Response
We sincerely thank both reviewers for their careful reading and insightful comments, which helped us improve the clarity, structure and scientific rigor of the manuscript. We deeply appreciate the time and effort dedicated to evaluating our work.
In response to the suggestions provided, we have revised the manuscript extensively, clarifying key methodological aspects, strengthening the Discussion and Conclusions, expanding the Introduction with broader literature context and refining the overall language and flow. We detail below how each point was addressed.
Reviewer 1:
- Title and Scope: The term "Horticultural Crops" is inaccurate since Zea mays and Hordeum vulgare are field crops rather than horticultural species. Suggest revising to "Agricultural Crops" or "Selected Crops" to avoid confusion.
R: we have changed for “….on the Invasive Plant Cenchrus setaceus and Crops”
- Abstract: The abstract overly emphasizes results on Cenchrus setaceus while neglecting quantitative findings on other species. Please include key results for non-target species.
We kept our original structure and focus on C. setaceus and added quantitative crop results from both growth chamber and greenhouse trials. We integrated key findings on crop selectivity and used phrasing like “varied by species and treatment” to reflect nuanced responses. Finally, we cleaned up sentence transitions for editorial quality.
- Keywords: Current keyword are too broad. Recommend replacing with more specific terms such as "allelopathic inhibition", "aqueous extracts", "seed germination", and "invasive species control".
R: changed
- Introduction: The introduction lacks background on related species with known allelopathic effects. Adding this would strengthen the rationale for studying A. thuscula and P. pendula.
R: we have revised the paragraphs and we have added depth into Artemisia for the A. thuscula rationale with artemisinin and essential oils allelopathy of Artemisia and for Plocama we underlined the studies on the anthraquinones on inhibition. (lines 115-133)
- Results Section: Avoid mechanically repeating numerical values from figures/tables. Instead, synthesize trends. Merge results from the two trials and discuss combined effects of high and low concentrations thematically.
R: We edited the preliminary trial text to avoid repetition and synthesize more, distinguishing concentration effects, which is thematically relevant. We maintained scientific precision without overwhelming readers with stats.
- Data Presentation: Replace raw data with standardized metrics: germination rate, germination energy, vigour index, plumule length, and radicle length. Move raw data to supplementary materials if necessary.
R: we have moved seed counts graphs to SI.
- Figure 5: The cluster analysis lacks biological interpretation and appears redundant. Recommend removal unless deeper ecological insights are provided.
R: at your recommendation, we have revised the decision and we moved the figure to Suppl Mat, giving some biological interpretation – possible trait explanations in the Discussion section (L 315-326), to make the clustering more meaningful. We would like to keep it as it provides a quick, intuitive sense of which species are similarly sensitive or robust.
- Discussion: Delete lengthy repetitions of results and speculative content on molecular mechanisms. Focus on practical implications of the findings (e.g., potential for integrated weed management) and limitations.
R: we carefully revised this section for coherence, flow and to reduce redundancy and avoid speculations. Now the version has a more elegant progression, as you suggested: findings -> practical implications -> integration into IWM -> environmental, social and economic aspects -> limitations -> future work.
- Methodology: The protocol for preparing extracts (fresh leaves chopped and soaked in water) deviates from classical allelopathy studies (typically using dried powdered material). Justify this approach with references and explain how it affects compound extraction efficiency.
R: Indeed many allelopathy studies use dried and powdered plant material because drying removes water and grinding increases extractable compound concentrations by disrupting cell structures. However, our choice of fresh leaves soaked in water was deliberate and ecologically justified. This method aims to simulate natural leaching events such as rainfall or dew dripping onto fresh foliage, where water soluble compounds are released into the environment. Fresh foliage lixiviates are widely used when the focus is on short-term release dynamics and ecological relevance rather than maximum extractable concentration (Thiebaut et al, 2018; Seifu et al 2023). Moreover, the drying process itself may alter chemical profiles by degrading heat-sensitive compounds or changing secondary metabolites balance (Vargas-Madriz et al 2023; Garcia et al 2021). For this reason, we followed protocols commonly applied in eco-physiological allelopathy studies, where fresh tissue and controlled soak conditions better reflect in situ interactions. We have now expanded Materials and Methods section accordingly, clarifying both the preparation method and its conceptual basis.
- Statistical Analysis: The description of statistical methods is confusing. Clarify how missing data or zeros were handled. Specify post-hoc tests used for pairwise comparisons.
R: we have revised the M & M section to improve clarity. Specifically, we now explain in more detail how zero values and missing data were treated for germination indices and seedling measurements. We have also specified the post-hoc tests used for pairwise comparisons (Dunn test with Bonferroni correction following Kruskal-Wallis, and Games-Howell following Welch-ANOVA). This ensures that the statistical workflow is fully transparent.
Minor Comments:
- Latin Names: Italicize all scientific names consistently in the main text and figures (e.g., Cenchrus setaceus, Artemisia thuscula).
R: done
- Table and Figure Captions: Captions should describe content directly (e.g., "Germination rates under different treatments") rather than analytical methods (e.g., "Cluster analysis based on FPG and GI").
R: changed
- Section Headings: Avoid informal headings like "Up-Scaled Trial". Use standard academic phrasing (e.g., "Greenhouse Validation Experiment").
R: changed
- Germination Conditions: Specify whether all species were germinated under identical conditions (temperature/light). If not, discuss potential impacts on results.
R: Within each biological repetition — both in the growth chamber and in the greenhouse — abiotic conditions were identical for all species and treatments. Each repetition included the full set of species exposed to the same environmental parameters concurrently. In the growth chamber, seeds were incubated under controlled conditions: a 12 h light/12 h dark cycle, temperatures of 25 °C (day) / 18 °C (night), relative humidity of 55–60%, and a light intensity of 50 μmol m⁻² s⁻¹ daily photon flux density.
In the greenhouse assays, species were incubated simultaneously within each repetition, but mean ambient temperatures differed slightly between months, as biological repetitions were distributed across February, March, and April. Recorded average day/night temperatures were:February: 21.18 °C / 18.08 °C; March: 22.14 °C / 12.91 °C; April: 25.33 °C / 15.10 °C. While these minor inter-month differences may have influenced absolute germination values, they affected all species equally within each assay. Therefore, comparisons across species and treatments within the same repetition remain valid, and differences between greenhouse and growth chamber assays are interpreted within this context. We added “potential impacts on results” in M&M: line 492 -> “..Within each biological repetition, all species were incubated under identical abiotic conditions, ensuring comparability across treatments. In the greenhouse assays, minor variation in ambient temperature occurred between the two repetitions due to seasonal progression (February to April), but both were conducted within the same climatic season (wet season), this limiting potential confounding effects.”
- Conclusion: Expand the conclusion to highlight practical applications and future work.
R: we have revised the conclusions and edited so that i) to avoid repeating results already detailed in discussion, ii) point to future work and iii) indicate practical applications.
- The manuscript requires extensive language polishing to address inconsistent tense usage, awkward phrasing, and grammatical errors for improved academic clarity.
R: We have polished nearly every section of the manuscript. We carefully checked and harmonized verb tenses and voice, clarified awkard sentence constructions, smoothed transitions, and corrected grammatical and semantic inconsistencies. In addition, we conducted a proofreading to ensure the style and tone meet the academic standards.
Reviewer 2 Report
Comments and Suggestions for Authors
The manuscrip ID 3836011 aims to evaluate the effect of water extracts from two native plants, Artemisia thuscula and Plocama pendula, on the germination and early growth of the invasive grass Cenchrus setaceus and several common crop species.
The manuscript addresses a highly relevant topic in plant allelopathy and weed management with potential application for controlling other invasive plant species and for managing other areas where C. setaceus is problematic.
In general, the manuscript is well written. However, several sections require substantial revision to improve clarity, organization, and consistency. The inclusion of the “Results” section before the “Material and Methods section” requires attention in the use of abbreviatures.
Starting with the Abstract, the methodology should be described more clearly. For example, the study was focused the use of two ratios of extracts 1:3 and 1:6, but in the first trial, the extract was diluted in two concentrations (1% and 10%) and this should be mentioned. Similarly, the "germination indices" mentioned in line 25 should be briefly defined.
The Introduction section is well structured and follows a logical sequence, providing a comprehensive overview of the study's scope and importance. the literature review should be expanded. For example, only one study on allelopathy and horticultural crops is cited [ref 9], which seems insufficient
The Results section requires major reorganization. Both the text and the tables/figures need revision. Values should be presented consistently, with the same number of decimal places. For example, between lines 129 and 140, values appear with one, two, or no decimal places. Units of all germination indices must be included in the text, tables, and figures. Results of statistical comparisons should be explicitly reported in the text.
Although the Discussion section includes some interpretation, it lacks sufficient contextualization with previous studies and should be substantially expanded The first paragraph, refers to: the date of seed collection and its implications on seed germination. But this was not mentioned in the Results section. Moreover, the sudy focuses on extract concentrations and dilutions, but there anot sufficiently discussed.
The Material and Methods section is confusing and should be rewritten for clarity. It is important to clarify the rationale for using two experiments (growth chamber and greenhouse), two extract concentrations (1:3 and 1:6), and dilutions (1% and 10%), and two germination periods (8 days and 14 days). In addition, the description of both experiments should be more precise, including the pH and electrical conductivity of all extracts (and dilutions). The integration of “two biological repetitions” in the analysis should be explained. It is unclear how many seedlings were measured per treatment. If I understand correctly, each growth chamber had five Petri dishes per treatment with 20 seeds each (total 100 seeds), yet you report measurements on 60 seedlings. Please clarify it. In lines 490-491 there is a reference to temperatures of February, March and April., but with no additional information. Why these temperatures are important?
The Conclusion synthetizes the main results of the study and is generally well written.
Other comments:
Line 20: Summarize the results of the preliminar trial.
Line 26: Indicate the duration of the growth chamber assays.
Line 32: The statement is too general; revise to better reflect the results.
Line 35: Why are “biologicals” and “IS” included as keywords?
Line 118: Specify which compounds are being referred to.
Line 129: Define “FPG”?
Lines 129-130: If FPG was 35% in treatment A and 52% in treatment B, how many seedlings were measured per treatment and per replicate?
Line 130-133: When p < 0.05, differences among treatments should be indicated using lowercase letters.
Line 134-135: Provide units for MGT and clarify the meaning of “Mean Germination Time index indicated similar ratio average values for treatments versus control”?
Line 137-141: Include units for “plumule and radicle length”.
Line 143: Clarify extract concentrations in the second assay, as was done in the preliminary trial (line 127). Include dilution percentages immediately after “concentrate” and add measurement units. The paragraph should be revised for clarity.
Line 151: The direct comparisons between the two sets of results should be made cautiously, considering initial differences in lixiviate concentrations. Revise it.
Line 164: Instead of “number of germinated seeds,” use germination percentage for Germination Dynamics.
Line 166: “C. setaceus” should be italicized.
Line 167: “H. vulgare” should be italicized.
Line 180: Clarify the meaning of “negative control”?
Line 208-216: Specify the number of seedlings measured per treatment, especially where FGP suggests fewer than 60 seedlings.
Line 233: “B. oleracea and R. sativus” should be italicized.
Line 292-286: The effect of “year of seed collection” is mentioned but not presented in the Results. Was this factor analyzed? Please clarify.
Line 487: Include the chemical composition of the substrate, including pH and electrical conductivity.
Line 593: review the entire listo f references. There are some inconsistencies.
Tables and Figures
Table 1: it shoud be formatted. Uniform decimal places and standardized abbreviations. Avoid using “C” in both treatments and concentration to prevent confusion. Consider replacing “Concentration” with “dilution”. Replace “na” with “0” where appropriate and use “-“ to indicate missing results. Review “MGT” values for C.s. species, Treatment “B” 100%. If the FPG=0, GI should be marked as “-“ or “na” and explained in a footnote. Standardize abbreviations for plumule and radicle lengths (“PL” and “RL” across all tables and figures).
Figure 1. The scientific names of the species should be italicized.
Figure 2. The scientific names of the species should be italicized. Include the meaning of “*” in the figure captions. Adjust graph sizes for better readability. In addition Standardize y-axes across graphs showing comparable data.
Table 2: Clarify whether values represent “germinated seeds” or “non-germinated seeds.
Figure 5. Needs major reformatting to improve readability.
Comments on the Quality of English LanguageThe manuscript is well written and only requires minor revisions.
Author Response
We sincerely thank both reviewers for their careful reading and insightful comments, which helped us improve the clarity, structure and scientific rigor of the manuscript. We deeply appreciate the time and effort dedicated to evaluating our work.
In response to the suggestions provided, we have revised the manuscript extensively, clarifying key methodological aspects, strengthening the Discussion and Conclusions, expanding the Introduction with broader literature context and refining the overall language and flow. We detail below how each point was addressed.
Reviewer 2
1. The inclusion of the “Results” section before the “Material and Methods section” requires attention in the use of abbreviatures.
R: we added the long form where needed
2. Starting with the Abstract, the methodology should be described more clearly. For example, the study was focused the use of two ratios of extracts 1:3 and 1:6, but in the first trial, the extract was diluted in two concentrations (1% and 10%) and this should be mentioned. Similarly, the "germination indices" mentioned in line 25 should be briefly defined.
R: We edited the abstract and explicitly mentioned 1% and 10% dilutions used in the preliminary trial. We also separated preliminary vs validation trial logic.
3. The Introduction section is well structured and follows a logical sequence, providing a comprehensive overview of the study's scope and importance. the literature review should be expanded. For example, only one study on allelopathy and horticultural crops is cited [ref 9], which seems insufficient.
R: we have edited according to your observations and expanded the topic: L 63-73; L 115-133
4. The Results section requires major reorganization. Both the text and the tables/figures need revision. Values should be presented consistently, with the same number of decimal places. For example, between lines 129 and 140, values appear with one, two, or no decimal places. Units of all germination indices must be included in the text, tables, and figures. Results of statistical comparisons should be explicitly reported in the text.
R: We have reorganized the first part of the Results section, condensing the preliminary trial narrative as recommended by Reviewer 1. Numerical values are now presented consistently with two decimal places in both the main text and table, except when values are zero. In the tables, all column headers have been updated to include the appropriate units: FPG (%), GI (index), PL & RL (mm), MGT (days). Units are also clearly indicated in all figure legends, and in the Results section, we now clarify the unit of each germination index upon first mention.
5. Although the Discussion section includes some interpretation, it lacks sufficient contextualization with previous studies and should be substantially expanded. The first paragraph, refers to: the date of seed collection and its implications on seed germination. But this was not mentioned in the Results section. Moreover, the study focuses on extract concentrations and dilutions, but there not sufficiently discussed.
R: As requested, we have expanded the first part of the Discussion section to improve contextualization with previous research. Regarding the date of seed collection, we clarify in the Materials and Methods that seeds for the preliminary and validation trials were collected in different months and years. Since this variable was not analysed statistically, we did not include it in the Results section. However, we agree that its potential influence on viability and dormancy deserves contextual reflection, which we have now developed in the revised Discussion. Additionally, we have improved the thematic interpretation of the concentration and dilution effects of the extracts, framing them in relation to plant physiological sensitivity and ecological application. Citations and comparisons with previous allelopathy studies have also been added to strengthen the discussion.
6. The Material and Methods section is confusing and should be rewritten for clarity. It is important to clarify the rationale for using two experiments (growth chamber and greenhouse), two extract concentrations (1:3 and 1:6), and dilutions (1% and 10%), and two germination periods (8 days and 14 days). In addition, the description of both experiments should be more precise, including the pH and electrical conductivity of all extracts (and dilutions). The integration of “two biological repetitions” in the analysis should be explained. It is unclear how many seedlings were measured per treatment. If I understand correctly, each growth chamber had five Petri dishes per treatment with 20 seeds each (total 100 seeds), yet you report measurements on 60 seedlings. Please clarify it. In lines 490-491 there is a reference to temperatures of February, March and April., but with no additional information. Why these temperatures are important?
R: The rationale for using two experimental systems (growth chamber and greenhouse) is now clearly stated as an effort to test the reproducibility and robustness of allelopathic effects under controlled versus semi-natural conditions.
The use of two extract ratios (1:3 and 1:6 w:v, leaves:water) corresponds to the preliminary and validation trials, respectively, and is now explicitly linked to the experimental phases.
Dilutions (1% and 10%) were only tested in the preliminary trials and are now clearly mentioned and justified where first introduced.
We have clarified that germination duration varied by species depending on known developmental patterns and has been harmonized into two groups (8-day vs. 14-day), now explained in both growth chamber and greenhouse assay descriptions.
Regarding biological repetitions, we confirm that two independent biological replicates were performed for each assay, and that abiotic conditions were maintained equal for all species within each repetition. Minor seasonal variation in greenhouse temperatures is acknowledged but confined to the same climatic season (wet season).
Regarding the number of seedlings measured we clarified: 60 randomly selected seedlings per treatment group were measured for plumule and radicle length for growth chamber assays in the validation trial, while all 100 were assessed in the preliminary trial. For both growth chamber and greenhouse assays, all germination events were counted (100 and 99 seeds, respectively), but seedling growth was not measured.
The inclusion of mean monthly greenhouse temperatures is intended to reassure readers about the environmental stability between the two biological repetitions; this point has been clarified in the revised version.
Lastly, while pH and electrical conductivity of the lixiviates were not measured directly, we ensured methodological consistency by using ultrapure water and uniform processing conditions across all extract preparations. The chemical properties of the greenhouse substrate (pH and salt content) are now reported.
Other comments:
7. Line 20: Summarize the results of the preliminary trial.
R: done
8-9. Line 26: Indicate the duration of the growth chamber assays. Line 32: The statement is too general; revise to better reflect the results.
R: we have rewritten the abstract.
10. Line 35: Why are “biologicals” and “IS” included as keywords?
The term “biologicals” is used here as a broad umbrella encompassing both microbiological agents and plant-derived natural products used in crop and weed management, as previously proposed (Collinge DB, Jensen DF, Rabiey M, Sarrocco S, Shaw MW, Shaw RH (2022) Biological control of plant diseases – what has been achieved and what is the direction? Plant Pathol 71:1024–1047. https://doi.org/10.1111/ppa.13555). Since our study evaluates natural lixiviates with potential bioherbicidal application, we consider “biologicals” an appropriate keyword to reflect the applied context of our findings.
The acronym “IS” refers to “invasive species”, with Cenchrus setaceus serving as a model organism in our assays. Given the global relevance of invasive species in both ecological and agricultural systems, we believe this keyword increases the manuscript’s discoverability for readers interested in invasion biology, restoration ecology, or plant-based suppression methods targeting invasive taxa. Yet, we modified the keywords, according to your observations and Reviewer’s 1 considerations: agroecology, biologicals, herbicide overuse, invasive species control, seed germination, aqueous extracts, allelopathic inhibition.
11. Line 118: Specify which compounds are being referred to.
R: Our preliminary assumption was that allelochemicals naturally present in aqueous extracts of A. thuscula and P. pendula would inhibit the germination of C. setaceus. Although the specific phytotoxic compounds in these species have not yet been identified, our hypothesis was based on their taxonomic proximity to known allelopathic plant groups and on ecological co-occurrence with crops. We further hypothesized that these compounds would exhibit low phytotoxicity toward horticultural species, considering the long-standing ecological and anthropogenic coexistence of these endemics with traditional agriculture in the Canary Islands. This assumption was supported by: i) the historical domestic and agricultural use of A. thuscula, and ii) the frequent presence of P. pendula near cultivated areas, suggesting potential compatibility.
12. Line 129: Define “FPG”?
R: done
13. Lines 129-130: If FPG was 35% in treatment A and 52% in treatment B, how many seedlings were measured per treatment and per replicate?
R: we have now clarified the number of samples taken into consideration for each calculus in Material and Methods.
14. Line 130-133: When p < 0.05, differences among treatments should be indicated using lowercase letters.
R: we have rewritten this part of the text; where relevant lowercase and uppercase letters are used (tables and figures).
15. Line 134-135: Provide units for MGT and clarify the meaning of “Mean Germination Time index indicated similar ratio average values for treatments versus control”?
R: we added units in text, figures and tables.
16. Line 137-141: Include units for “plumule and radicle length”.
R: mm added
17. Line 143: Clarify extract concentrations in the second assay, as was done in the preliminary trial (line 127). Include dilution percentages immediately after “concentrate” and add measurement units. The paragraph should be revised for clarity.
R: we have completely edited the first paragraphs from the Results section as required by your comments and Reviewer’s 1.
18. Line 151: The direct comparisons between the two sets of results should be made cautiously, considering initial differences in lixiviate concentrations. Revise it.
R: Currently we only compare only within “preliminary trial” or “validation trial”
19. Line 164: Instead of “number of germinated seeds,” use germination percentage for Germination Dynamics.
R: we have moved seed counts into SI.
20. Line 166: “C. setaceus” should be italicized.
R: done
21. Line 167: “H. vulgare” should be italicized.
R: done
22. Line 180: Clarify the meaning of “negative control”?
R: we refer to the not-treated group; in pathogenic assays we have “a negative” control, where nothing is applied and “positive” control, where the pathogen is applied. Here it was a simple misspelling error, given our background. Therefore we deleted “negative”.
23. Line 208-216: Specify the number of seedlings measured per treatment, especially where FGP suggests fewer than 60 seedlings.
R: we have now added exactly how many seeds and seedlings were used per category of assay (growth chamber vs greenhouse) and trial (preliminary vs validation).
24. Line 233: “B. oleracea and R. sativus” should be italicized.
R: done
25. Line 292-286: The effect of “year of seed collection” is mentioned but not presented in the Results. Was this factor analysed? Please clarify.
Indeed, the timing of seed collection was not previously mentioned, and we have now added this information in the Materials and Methods section (Line 468). Seeds for the preliminary trial were collected at two time points within the same year (September and December), both falling within the regional wet season (October to May). For the validation trial, seeds were collected two years later in September. The year or month of collection was not analysed as an experimental factor, as doing so would have introduced additional complexity with too few replicates for meaningful inference. Moreover, the preliminary and validation trials were not statistically compared across years. The observation on germination variability is presented in the Discussion only as contextual support, not as a tested variable, and the paragraph has been edited to reflect its descriptive (not inferential) nature (line 277).
26. Line 487: Include the chemical composition of the substrate, including pH and electrical conductivity.
We have updated the Materials and Methods section (Line 487) to include all available physicochemical parameters of the commercial substrate used (COMPO Sana® Universal). While electrical conductivity was not reported by the manufacturer, we now specify the product’s salt content (<3 g L⁻¹) as a relevant indicator. This clarification has been added to the revised manuscript (line 486).
27. Line 593: review the entire list of references. There are some inconsistencies.
R: done
Tables and Figures
28. Table 1: it should be formatted. Uniform decimal places and standardized abbreviations. Avoid using “C” in both treatments and concentration to prevent confusion. Consider replacing “Concentration” with “dilution”. Replace “na” with “0” where appropriate and use “-“ to indicate missing results. Review “MGT” values for C.s. species, Treatment “B” 100%. If the FPG=0, GI should be marked as “-“ or “na” and explained in a footnote. Standardize abbreviations for plumule and radicle lengths (“PL” and “RL” across all tables and figures).
R: we have applied the following changes:
All values have been standardized to two decimal places across all tables, including those in the Supplementary Material. Abbreviations have been fully standardized as follows: A = Artemisia thuscula, B = Plocama pendula, C = control, PL = plumule length, RL = radicle length. These now match the formatting used throughout the manuscript. We used “Conc” to indicate concentration as it correctly reflects the % dilution of the original extract (1%, 10%, 100%) in volume terms and is scientifically accurate in the context of aqueous extract assays. The label “na” has not been replaced with “0”, since this would incorrectly suggest that measurements were performed and returned a zero value. In cases where no germination occurred (FPG = 0), it was not possible to obtain GI, PL, or RL values. These cells are now marked as “–” to indicate non-applicable/missing results, and this is clearly explained in a revised table footnote.
The MGT values for C. setaceus, Treatment B (100%) have been reviewed and corrected where necessary. The table footnote has been expanded to clearly explain the meaning of “–” values, specifically when indices could not be calculated due to lack of germination.
29. Figure 1. The scientific names of the species should be italicized.
R: done
30. Figure 2. The scientific names of the species should be italicized. Include the meaning of “*” in the figure captions. Adjust graph sizes for better readability. In addition, Standardize y-axes across graphs showing comparable data.
R: Regarding the meaning of “*” in the figure captions: there must have been a misunderstanding – we did not use this symbol in figures; what you might have seen similar to it are the data outliers showed by the boxplots. We have uploaded the files separately and the editorial team will introduce the graphics adequately in text; we mention that the sizes correspond to instructions for authors. We have standardized y axes for both figures in the manuscript.
31. Table 2: Clarify whether values represent “germinated seeds” or “non-germinated seeds.
R: We changed for “non-germinated”
32. Figure 5. Needs major reformatting to improve readability.
R: we have moved the figure to SM according to the recommendation of Reviewer 1.
Round 2
Reviewer 1 Report
Comments and Suggestions for Authors
I have reviewed the authors' responses to the initial review comments and the revised manuscript. The authors have provided a comprehensive and serious response to all the feedback. All raised issues have been adequately addressed, and the revisions are well integrated into the manuscript.
The overall quality of the manuscript has been significantly improved. I recommend acceptance for publication.
A minor note on Table 1: Its descriptive caption should follow the table, not the title.
Author Response
Dear Reviewer, we have changed the title of the table. This preserves all the information but formats it exactly as you suggested.
Thank you for your time and interest!
Reviewer 2 Report
Comments and Suggestions for Authors
The Authors have successfully addressed my concerns in this resubmitted version of the manuscript. The overall quality looks much improved. Only minor editorial adjustments are needed before the manuscript can be considered suitable for publication in “Plants”.
Minor comments:
Line 66: The abbreviation “IWM” is introduced for the first time,and afterwards only the abbreviation should be used. Please revise line 299 accordingly.
Line 78: Add a period after “et al”.
Line 79: Provide the genus name for A. absinthium and A. vulgaris, because these species were not mentioned previously.
Line 132: Revise the sentence referring to the genus “Plocama” and its family “Rubiaceae” for clarity.
Line 231: Consider using “L. sativa”, “Z. mays”, and “C. setaceus”
Tables and Figures
Table 1: Replace “C.s.” and “L.s.” with “C. setaceus” and “L. sativa”, respectively, to ensure consistency. In the “Conc”, indicate “0”.
Figures 1 and 2. Add“FFG (%)”, “GI”, and “MGT (day)” to the “yy” axis, for clarity. Include the meaning of the columns and bars.
Author Response
Dear Reviewer,
please find below the modified parts as you required.
Line 66: The abbreviation “IWM” is introduced for the first time, and afterwards only the abbreviation should be used. Please revise line 299 accordingly.
R: All Integrated Weed Management mentions following the first have been changed to “IWM”.
Line 78: Add a period after “et al”.
R: Period added.
Line 79: Provide the genus name for A. absinthium and A. vulgaris, because these species were not mentioned previously.
R: Added for both species, and for Artmisia argyi in line 81 for the same reason.
Line 132: Revise the sentence referring to the genus Plocama and its family Rubiaceae for clarity.
R: Sentence changed
Line 231: Consider using “L. sativa”, “Z. mays”, and “C. setaceus”.
R: Species named abbreviated.
Tables and Figures
Table 1: Replace “C.s.” and “L.s.” with “C. setaceus” and “L. sativa”, respectively, to ensure consistency. In the “Conc”, indicate “0”.
Figures 1 and 2: Add “FPG (%)”, “GI”, and “MGT (day)” to the “yy” axis, for clarity. Include the meaning of the columns and bars.
R: Text was modified within the figure and legends are self-explanatory now.
The authors thank you for your time and effort.